# Microbiome Migration from Soil to Leaves in Maize and Rice

**DOI:** 10.3390/microorganisms13040947

**Published:** 2025-04-20

**Authors:** Jiejia Ma, Qianze Peng, Silu Chen, Zhuoxin Liu, Weixing Zhang, Chi Zhang, Xiaohua Du, Shue Sun, Weiye Peng, Ziling Lei, Limei Zhang, Pin Su, Deyong Zhang, Yong Liu

**Affiliations:** 1Longping Branch, College of Biology, Hunan University, Changsha 410082, China; mjj1999@hnu.edu.cn (J.M.); chensilu@hnu.edu.cn (S.C.); xin99@hnu.edu.cn (Z.L.); lei121@hnu.edu.cn (Z.L.); zhangdeyong@hhrrc.ac.cn (D.Z.); 2State Key Laboratory of Hybrid Rice and Institute of Plant Protection, Academy of Agricultural Sciences, Changsha 410125, China; pengqianze@hainanu.edu.cn (Q.P.); 17793687462@163.com (W.Z.); 15580076854@163.com (W.P.); z13933758941@163.com (L.Z.)

**Keywords:** plant microbiome, soil microbiome, synthetic microbial community, rice, maize, Burkholderiales

## Abstract

The interactions between plants and microbes are essential for enhancing crop productivity. However, the mechanisms underlying host-specific microbiome migration and functional assembly remain poorly understood. In this study, microbiome migration from soil to leaves in rice (*Oryza sativa*) and maize (*Zea mays*) was analyzed through 16S rRNA sequencing and phenotypic assessments. When we used the same soil microbiome source to grow rice and maize, microbiota and functional traits were specifically enriched by maize in its phyllosphere and rhizosphere. This indicated that plants can selectively assemble microbiomes from a shared microbiota source. Therefore, 22 strains were isolated from the phyllospheres of rice and maize and used to construct a synthetic microbial community (SynCom). When the soil for rice and maize growth was inoculated with the SynCom, strains belonging to *Bacillus* were enriched in the maize phyllosphere compared to the rice phyllosphere. Additionally, a strain belonging to *Rhizobium* was enriched in the maize rhizosphere compared to the rice rhizosphere. These results suggest that plant species influence the migration of microbiota within their respective compartments. Compared with mock inoculation, SynCom inoculation significantly enhanced plant growth. When we compared the microbiomes, strains belonging to *Achromobacter*, which were assembled by both rice and maize, played a role in enhancing plant growth. Our findings underscore the importance of microbial migration dynamics and functional assembly in leveraging plant–microbe interactions for sustainable agriculture.

## 1. Introduction

In natural environments, both the surface [1] and interior [2] of all plants are colonized by complex and diverse microbial communities, comprising bacteria, fungi, archaea, viruses, and protists. These microbial communities are referred to as the plant microbiome [3]. The plant microbiome includes the rhizosphere microbiome, phyllosphere microbiome, and endophytic microbiome. It plays a crucial role in promoting plant growth [4,5,6,7], enhancing disease resistance [8], facilitating nutrient uptake [9,10], and increasing abiotic stress resistance [11]. The assembly of the plant microbiome is influenced by many factors, including the plant species, plant genotype [12,13,14], ecological environment [15,16], agricultural practices [17], microbial source [18], and plant–microbiota interaction [19]. Soil is the main source of the plant microbiome. A study on soil exchange showed that soil microorganisms colonize the plant phyllosphere [20]. The overlap in microbial species composition among the soil, rhizosphere, and phyllosphere microbiomes also suggests that soil is the source of plant microorganisms [21,22]. Studies have demonstrated that plant rhizospheres secrete specific compounds [23], including flavonoids, citric acid, malic acid [24,25], and coumarins [26], to recruit soil microorganisms. Following recruitment into the rhizosphere, a subset of these microorganisms may colonize the phyllosphere under the selective pressure imposed by the plant host [27]. However, the process of how soil microbiomes migrate to become plant microbiomes remains to be elucidated.

Rice and maize, the world’s most important staple foods [28], provide direct food sources for billions of people. Maize is also a vital resource in global agriculture, serving as a cornerstone for both livestock nutrition and various industrial processes [29], with high nutritional and economic value. During their growth, rice and maize are challenged by bacterial [30] and fungal [31] diseases, as well as by abiotic stresses like drought and salinity. These challenges reduce their yield. Therefore, it is vital to promote the growth of rice and maize and enhance their biotic and abiotic stress resistance. A recent study showed that beneficial microorganisms can induce systemic resistance in plants, thereby increasing plants’ disease resistance [32]. For instance, *Streptomyces* sp. SS8, a Plant-Growth-Promoting Rhizobacterium (PGPR), induces systemic resistance in rice, enhancing its resistance to bacterial leaf spot and promoting its growth [33]. The disease resistance and growth of maize can both be enhanced by reshaping the structure of the phyllosphere microbiome [34]. Some specific microorganisms in the roots of maize improve plant growth and resistance to salt stress [35]. These research findings further underscore the importance of the plant microbiome in promoting plant growth and enhancing resistance to abiotic stresses. Nevertheless, these beneficial functions for rice and maize are largely dependent on the specific assembly of the plant microbiome [36,37,38,39]. At present, studies on the microbiome assembly of rice and maize mainly focus on the host, environment, and microbe–microbe interactions [36,40,41,42,43,44]. Soil serves as the primary source of the plant microbiome, and studies in *Arabidopsis thaliana* have demonstrated that soil-derived microbial communities can migrate to the phyllosphere [27]. However, research on the migration of soil microbiomes to the rhizosphere and phyllosphere of rice and maize is limited.

A synthetic microbial community (SynCom) is a simplified microbial community characterized by high efficiency and controllability. It is formed by mixing various microbial strains with known functions and clear taxonomic status in specific proportions under controlled conditions [45]. In agriculture, a SynCom can be utilized to develop biofertilizers [46], improve crop yields [47], and enhance plant stress resistance [48]. Zhang and colleagues (2019) found the inoculation of rice with a SynCom under conditions of rich organic nitrogen promoted its growth [49]. Huang and colleagues (2023) revealed that *Botrytis cinerea* proliferation can be prevented by using a simplified microbial community [50]. Esau De la Vega-Camarillo and colleagues (2023) found that the assembly of a SynCom with Plant-Growth-Promoting Bacteria (PGPB) can promote the growth of maize and reduce the probability of maize infection with rust fungus [51]. Zhou and colleagues (2022) showed that a SynCom can be used against soil-borne fusarium wilt in tomatoes [52]. A SynCom of the plant core microbiome can be a potential biological control tool [53]. Therefore, using a SynCom to study the migration of soil microbiomes to plant rhizospheres and phyllospheres is a viable strategy.

In order to characterize the migration patterns of the soil microbiome towards rice and maize, we first analyzed the microbiomes present in the rhizosphere and phyllosphere of both crops grown in the same soil. We then isolated and purified the phyllosphere microbiota of rice and maize after migration, and we constructed a SynCom from these isolates. Finally, the SynCom was used to validate the characteristics of soil microbiome migration towards rice and maize and to reveal the reasons behind growth promotion in these plants after migration. This study provides a method and insight for the development of microbiome-based products aimed at increasing crop yields and promoting sustainable agricultural development.

## 2. Materials and Methods

### 2.1. Plant Materials, Seed Germination, and Growth Conditions

The plant experimental materials were rice (Zhonghua 11) and maize (Zhengdan 958). Rice and maize seeds were dried in an oven for 12 h, placed in a beaker, and soaked in sterile water. The seeds floating on the surface were removed. The sterilization operation was performed as follows: the cleaned seeds were transferred to a sterile conical flask inside a clean bench, soaked in a 10% sodium hypochlorite solution for 15 min, washed with sterile water, and then re-soaked in sterile water. The above surface sterilization steps were repeated after 12 h. The seeds were placed in a refrigerator at 4 °C for overnight treatment, then placed for germination in a culture room at 25 °C with 70% relative humidity and a light/dark photoperiod of 14 h:10 h. The germinated seeds were sown and cultured in a culture chamber at 28 °C with 70% relative humidity and a light/dark photoperiod of 14 h:10 h. The soil preparation method was consistent with previous research methods [54], using a field soil filtrate collected from Changsha, China (28°11′44.963″ N, 113°05′1.684″ E), mixed with growth soil (PINDSTRUP substrate, Syddjurs, Denmark; autoclaved twice) to obtain the cultivation soil. The germinated seeds of both maize and rice were planted in the same batch of prepared soil.

### 2.2. Sample Collection

After 14 days of maize growth and 18 days of rice growth, the plants were collected. The roots (with the gentle shaking of the samples to dislodge loose soil particles) and leaves were separately placed into 50 mL sterile centrifuge tubes, and care was taken to prevent cross-contamination during sampling. A 30 mL volume of 1× PBS buffer was added to the leaf samples to extract the phyllosphere microbiome, and 20 mL of 1× PBS buffer was added to the root samples to extract the rhizosphere microbiome. The samples were shaken at 30 °C at 200 rpm for 30 min, followed by ultrasonication at 30 kHz for 10 min. After sonication, the samples were centrifuged at 19,000× *g* and 4 °C for 20 min. The supernatant was removed, then 1.6 mL of sterile water was added to resuspend the extracted microbiome. The samples were stored in a refrigerator at −80 °C for later use. For each experimental treatment group, at least three biological replicates were conducted.

### 2.3. DNA Extraction, PCR Amplification, and Sequencing

The total DNA was extracted using the MagPure Soil DNA LQ Kit (Magen, Shanghai, China) according to the manufacturer’s instructions. The quality and quantity of the DNA were verified using a NanoDrop ND-1000 spectrophotometer (Thermo Fisher Scientific, Waltham, MA, USA) and agarose gel electrophoresis, respectively. The extracted DNA was diluted to a concentration of 1 ng/μL and stored at −20 °C until further processing. The PCR amplification of bacterial 16S rRNA gene fragments (V3-V4 region) was performed using Takara Ex Taq (Takara, Beijing, China) and the primers 343F (5′-TACGGRAGGCAGCAG-3′) and 798R (5′-AGGGTATCTAATCCT-3′). Amplicons were visualized using agarose gel electrophoresis and purified twice using Agencourt AMPure XP beads (Beckman Coulter, Pasadena, CA, USA) twice. After purification, the DNA was quantified using the Qubit dsDNA assay kit (Yeasen, Shanghai, China). Equal amounts of purified DNA were pooled for sequencing on the NovaSeq 6000 platform (Illumina Inc., San Diego, CA, USA) at Shanghai OEbiotech (Shanghai, China).

### 2.4. Bioinformatics Analysis

The 16S rRNA gene fragment sequences were processed using vsearch v.2.22.1 [55]. All OTUs were annotated using the Silva v138.1 reference database [56]. Bacterial functional profiles were predicted using Functional Annotation of Prokaryotic Taxa (FAPROTAX) [57]. Animal parasites or symbionts and their taxa were generated through FAPROTAX. Alpha diversity analysis was carried out using the EasyAmplicon script [58]. Unconstrained and constrained principal coordinate analyses (PCoA and CPCoA) based on Bray–Curtis distances were performed using the R package amplicon [58]. Analyses of differential species and function abundance and determinations of significantly differently abundant taxa and functions were performed using the unpaired two-tailed Student’s *t*-test. The relevant data were visualized by using the R package ggplot2 v.3.5.2.

### 2.5. Isolation of Bacteria

The leaves of 14-day-old maize and 18-day-old rice were mixed and ground into a homogenate using sterile water. The extracted solution was serially diluted tenfold with sterile water to achieve dilution factors of up to 10^6^. Then, 90 μL of the diluted sample was inoculated onto nutrient agar (NA) plates. Single colonies with different colors and morphologies were individually activated overnight at 28 °C and 200 rpm for activation culture. The activated bacterial suspensions were streaked onto NA plates and cultured upside down at 28 °C, and observations were made within 24 to 48 h to determine whether single bacterial colonies had formed. Bacteria were purified by means of continuous streaking until pure cultures were obtained. Each single bacterial colony was reactivated, mixed with glycerol, and stored in a refrigerator at −80 °C for subsequent species identification and sequencing. Cultivated bacteria were identified by means of Sanger sequencing using the primers 27F (5′-AGAGTTTGATCCTGGCTCAG-3′) and 1492R (5′-TACGGCTACCTTGTTACGACTT-3′).

### 2.6. Synthetic Community Assembly and Inoculation

The SynCom assembly was consistent with previous research methods [59]. The isolated strains were individually cultured overnight in liquid media at 28 °C and 200 rpm to activate the bacterial strains. The activated strains were then streaked onto NA plates and incubated at 28 °C until colonies appeared. A single colony from each strain was selected and resuspended in 1 mL 10 mM MgCl_2_ solution. The resuspended strains were vortexed for 3 min and then sonicated for 45 s. This operation was repeated twice. Equal volumes of the individual strain suspensions were mixed to create the final SynCom solution. The concentration of the final SynCom solution was adjusted to OD_600_ = 0.02 using 10 mM MgCl_2_ solution, and it was then aliquoted for immediate use and stored in a refrigerator at −80 °C for subsequent sequencing.

The growth soil (PINDSTRUP substrate, Syddjurs, Denmark) was autoclaved three times, with each session spaced six hours apart. The sterilized soil was then mixed with the SynCom solution at a 1:1 ratio (*w*/*v*). For the control group, an equal volume of 10 mM MgCl_2_ was used instead of the SynCom solution. Seeds were surface-sterilized using the method described previously and sown in the prepared soil. The plants were grown in a greenhouse maintained at 28 °C. During the growth period, 1/2 MS solution was applied weekly to nourish the plants. Leaves and roots were collected from the maize plants after 14 days and from the rice plants after 18 days. The phyllosphere and rhizosphere microbiomes were extracted from these samples and then stored at −80 °C for further analysis of the bacterial community composition. Agronomic traits such as the shoot length, shoot weight, plant fresh weight, and plant length were measured for both the control (CK) and treatment groups.

## 3. Results

### 3.1. The Pattern of Microbiome Migration Varies Between Rice and Maize

The microbiome communities showed differences when the same soil was used to grow rice and maize. The constrained principal coordinate analysis (CPCoA) showed the clear separation of the soil, rhizosphere, and phyllosphere microbial communities for rice (Figure 1B; *p* = 0.001, 53.6% of variance, PERMANOVA). Moreover, the alpha diversity of the rice microbiomes progressively and significantly decreased from the soil to the rhizosphere and then to the phyllosphere of rice (Figure 1C). Analysis of the microbiome compositions found that the predominant bacterial orders in the soil, rhizosphere, and phyllosphere were Bacteroidales (with relative abundances of 0.40%, 3.06%, and 33.12% for the soil, rhizosphere, and phyllosphere, respectively), Burkholderiales (6.96%, 45.03%, 3.96%), Clostridiales (0.27%, 0.81%, 23.62%), Xanthomonadales (37.42%, 1.17%, 0.02%), Enterobacteriales (0.44%, 0.19%, 11.87%), Rhizobiales (15.51%, 8.36%, 3.90%), Acidobacteriales (10.40%, 12.22%, 0.01%), Lactobacillales (0.03%, 0.29%, 8.15%), Sphingobacteriales (5.68%, 2.31%, 0.85%), Rhodospirillales (3.89%, 2.91%, 0.31%), Caulobacterales (0.12%, 2.01%, 1.26%), and Pseudomonadales (0.03%, 2.53%, 0.65%) (Figure 1A). The analysis of variance (ANOVA) revealed that, compared to those in soil, the relative abundances of Burkholderiales and Caulobacterales in the rice rhizosphere microbiome were significantly higher, whereas the relative abundances of Xanthomonadales, Rhizobiales, Sphingobacteriales, and Rhodospirillales were significantly lower (Figure 1A). Additionally, compared to those in the soil, the relative abundances of Bacteroidales, Clostridiales, Pseudomonadales, Lactobacillales, and Enterobacteriales in the rice phyllosphere were significantly higher. Conversely, Burkholderiales, Sphingobacteriales, Xanthomonadales, Rhizobiales, Acidobacteriales, and Rhodospirillales showed significantly lower relative abundances in the rice phyllosphere compared to the soil (Figure 1A). These results indicate that the soil microbiome, serving as a source for the plant microbiota, possesses a rich species composition and shows significant variation after migrating to the rice rhizosphere and phyllosphere.

For maize, CPCoA and PERMANOVA showed a clear separation of microbial communities between different compartments of maize (Figure 2B; *p* = 0.001, 62.6% of variance). A Shannon index analysis also indicated that the microbial diversity in the soil was significantly different from that in the maize phyllosphere and rhizosphere (Figure 2C). The predominant bacterial orders in the soil, rhizosphere, and phyllosphere were Burkholderiales (with relative abundances of 0.40%, 3.06%, and 33.12% for the soil, rhizosphere, and phyllosphere, respectively), Sphingomonadales (0.62%, 0.56%, 37.03%), Xanthomonadales (37.42%, 9.10%, 0.05%), Bacteroidales (0.40%, 0.36%, 19.76%), Rhizobiales (15.51%, 9.07%, 2.43%), Acidobacteriales (10.40%, 12.28%, 0.03%), Clostridiales (0.27%, 0.21%, 15.34%), Enterobacteriales (0.44%, 0.45%, 9.89%), Sphingobacteriales (5.68%, 2.84%, 0.06%), Rhodospirillales (3.89%, 3.68%, 0.05%), Pseudomonadales (0.03%, 0.16%, 3.31%) and Lactobacillales (0.03%, 0.05%, 2.13%) (Figure 2A). Compared with that in the soil, the relative abundance of Burkholderiales was significantly higher in the maize rhizosphere microbiome, whereas the relative abundances of Xanthomonadales, Rhizobiales, and Sphingobacteriales were significantly lower (Figure 2A). Additionally, in the maize phyllosphere microbiome, the relative abundances of Sphingomonadales and Bacteroidales were significantly higher than in the soil, while those of Xanthomonadales, Rhizobiales, Acidobacteriales, Sphingobacteriales, and Rhodospirillales were significantly lower (Figure 2A). These results indicate that maize and rice share the same soil microbial sources, and the soil microbiome exhibits similarities during its migration to their respective rhizosphere and phyllosphere microbiomes. For instance, there is an enrichment of Burkholderiales in the rhizosphere and Bacteroidales in the phyllosphere for both crops. However, significant differences are also observed, such as the significant enrichment of Clostridiales, Pseudomonadales, Lactobacillales, and Enterobacteriales in the rice phyllosphere, and a significant enrichment of Sphingobacteriales in the maize phyllosphere.

### 3.2. The Pattern of Microbiome Function Migration Varies Between Rice and Maize

To assess the impact of predicted functional composition migration on rhizosphere and phyllosphere microbiomes in rice, PERMANOVA and CPCOA ordinations were implemented. They indicated that microbiome migration explained 55.5% of the variation in the functional composition of the soil, rhizosphere, and phyllosphere (Figure 3B). In rice, the primary functional composition of the microbial community was chemoheterotrophy (with relative abundances of 41.21%, 45.78%, and 31.08% for the soil, rhizosphere, and phyllosphere, respectively), aerobic_chemoheterotrophy (39.27%, 44.58%, 4.30%), fermentation (1.56%, 1.04%, 26.66%), animal_parasites_or_symbionts (1.33%, 2.05%, 6.50%), nitrate_reduction (0.66%, 0.27%, 5.43%), human_associated (0.61%, 0.30%, 4.92%), mammal_gut (0.12%, 0.17%, 4.75%), human_gut (0.12%, 0.16%, 4.75%), aromatic_compound_degradation (7.24%, 2.11%, 0.08%), plant_pathogen (1.19%, 0.60%, 1.94%), arsenite_oxidation_detoxification (0.01%, 0.14%, 1.66%), dissimilatory_arsenite_oxidation (0.01%, 0.14%, 1.66%), and nitrate_respiration (0.52%, 0.08%, 1.06%) (Figure 3A). The analysis of variance (ANOVA) revealed that the relative abundances of chemoheterotrophy and aerobic_chemoheterotrophy in the rice rhizosphere were significantly higher than those in the soil. Conversely, the relative abundances of human_associated, nitrate_reduction, aromatic_compound_degradation, and nitrate_respiration were significantly lower in the rice rhizosphere compared to the soil (Figure 3A). Additionally, the analysis found that the relative abundances of fermentation, animal_parasites_or_symbionts, nitrate_reduction, human_gut, mammal_gut, human_associated, arsenite_oxidation_detoxification, and dissimilatory_arsenite_oxidation in the rice phyllosphere were significantly higher than those in the soil. Conversely, the relative abundances of chemoheterotrophy, aromatic_compound_degradation, and aerobic_chemoheterotrophy were significantly lower in the rice phyllosphere compared to the soil (Figure 3A).

For maize, the PERMANOVA and CPCOA ordinations indicated that microbiome migration explained 51.1% of the variation in the functional composition of the soil, rhizosphere, and phyllosphere (Figure 4B). In maize, the primary functional composition across different compartments was chemoheterotrophy (with relative abundances of 41.21%, 43.92%, and 36.02% for the soil, rhizosphere, and phyllosphere, respectively), aerobic_chemoheterotrophy (39.27%, 42.17%, 18.98%), fermentation (1.56%, 1.16%, 16.31%), animal_parasites_or_symbionts (1.33%, 1.73%, 4.59%), aromatic_compound_degradation (7.24%, 0.70%, 1.32%), nitrate_reduction (0.66%, 0.83%, 4.23%), human_associated (0.61%, 0.33%, 4.45%), human_gut (0.12%, 0.16%, 4.37%), mammal_gut (0.12%, 0.16%, 4.37%), plant_pathogen (1.19%, 2.20%, 0.57%), cellulolysis (2.92%, 0.43%, 0.01%), nitrogen_fixation (0.16%, 1.72%, 0.01%), ureolysis (0.03%, 0.66%, 0.72%), and hydrocarbon_degradation (0.18%, 0.70%, 0.56%) (Figure 4A). Compared to those in the soil, the relative abundances of ureolysis, nitrogen_fixation, chemoheterotrophy, hydrocarbon_degradation, and aerobic_chemoheterotrophy in the maize rhizosphere were significantly higher. Conversely, the relative abundances of human_associated and aromatic_compound_degradation were significantly lower in the maize rhizosphere compared to the soil (Figure 4A). Additionally, in the maize phyllosphere, the relative abundances of fermentation, human_gut, mammal_gut, nitrate_reduction, human_associated, and animal_parasites_or_symbionts were significantly higher than those in the soil, while the relative abundances of aerobic_chemoheterotrophy, aromatic_compound_degradation, and nitrogen_fixation were significantly lower (Figure 4A). In summary, the functions of the soil microbiome were significantly changed as it migrated towards the rhizosphere and phyllosphere of maize and rice, and it exhibited marked differences in its migration processes towards either maize or rice.

### 3.3. Isolation of Microbiota from the Phyllosphere After Migration Events

In order to further investigate the migration events for rice and maize, we isolated and purified phyllosphere microbiota from both rice and maize plants. A total of 22 strains were isolated, purified, and identified based on differences in their morphology and 16S rRNA sequencing (Figure 5A). The 22 strains were classified into two phyla (Firmicutes and Proteobacteria), eight orders (Bacillales, Burkholderiales, Enterobacteriales, Neisseriales, Pseudomonadales, Rhizobiales, Sphingomonadales, and Xanthomonadales), and eleven genera (*Bacillus*, *Achromobacter*, *Escherichia-Shigella*, *Pantoea*, *Serratia*, *Chromobacterium*, *Acinetobacter*, *Pseudomonas*, *Bradyrhizobium*, *Sphingobium* and *Xanthomonas*). The proportions of the 22 strains from the different orders were as follows: Pseudomonadales accounted for 36.36%; Enterobacteriales accounted for 18.18%; Bacillales and Burkholderiales each accounted for 13.63%; and Neisseriales, Rhizobiales, Sphingomonadales, and Xanthomonadales each accounted for 4.55% (Figure 5B). Besides the strains from Neisseriales and Bacillales, the other strains belong to the top 10 most abundant bacterial orders in the phyllosphere of rice and maize. These 22 strains isolated from the phyllosphere of rice and maize are closely related to plant functions and can be used to simulate the composition of the phyllosphere microbiome (Appendix A).

### 3.4. Investigatione of SynCom Migration to Rhizosphere and Phyllosphere of Rice and Maize

The 22 strains isolated and purified from the phyllospheres of rice and maize were used to prepare a synthetic microbial community (SynCom) and were inoculated into the soil for plant growth. Here, the top 20 most abundant OTUs in the SynCom were OTU_4 (relative abundance of 17.6% in the SynCom), OTU_5 (13.87%), OTU_3 (10.75%), OTU_2 (8.46%), OTU_8 (7.52%), OTU_7 (6.55%), OTU_9 (5.38%), OTU_14 (5.34%), OTU_13 (5.05%), OTU_12 (4.54%), OTU_28 (2.51%), OTU_17 (2.34%), OTU_1 (1.88%), OTU_18 (1.76%), OTU_6 (0.94%), OTU_254 (0.77%), OTU_15 (0.32%), OTU_45 (0.16%), OTU_20 (0.14%), and OTU_30 (0.12%) (Figure 6). These OTUs are classified into the following orders: Bacillales (OTU_14, OTU_12, OTU_45), Bacteroidales (OTU_15, OTU_20, OTU_30), Burkholderiales (OTU_9), Enterobacteriales (OTU_1, OTU_3, OTU_4), Neisseriales (OTU_5), Pseudomonadales (OTU_6, OTU_254, OTU_8, OTU_7, OTU_13, OTU_28), Rhizobiales (OTU_18), Sphingomonadales (OTU_2), and Xanthomonadales (OTU_17) (Figure 6). The microbiome composition analysis demonstrated that the top 20 most abundant OTUs in the SynCom could be detected in both the rhizosphere and the phyllosphere, indicating microbial migration from the SynCom, applied through soil inoculation, to various plant parts during growth (Figure 6). The 16S rRNA sequences of these OTUs were compared with those of the strains in the SynCom. This analysis confirmed that *Bacillus* (OTU_14, OTU_12, OTU_45), *Achromobacter* (OTU_9), *Escherichia-Shigella* (OTU_4), *Pantoea* (OTU_1), *Serratia* (OTU_3), *Chromobacterium* (OTU_5), *Acinetobacter* (OTU_8, OTU_6), *Pseudomonas* (OTU_7, OTU_28, OTU_254, OTU_13), *Rhizobium* (OTU_18), *Sphingobium* (OTU_2), and *Xanthomonas* (OTU_17) originated from the isolated strains (Appendix A). These results indicate that most of the strains in the SynCom can migrate from the soil to the plant rhizosphere and phyllosphere.

An analysis of the migration of species from the SynCom to the rice rhizosphere and phyllosphere revealed that, as compared to the SynCom, OTU_1 (Enterobacteriales) and OTU_6 (Pseudomonadales) were significantly enriched in the rice rhizosphere, while OTU_254 (Pseudomonadales) was significantly enriched in the rice phyllosphere (Figure 6A). A further comparative analysis between the rice phyllosphere and rhizosphere showed that *Pseudomonas* (OTU_254) and OTU_4 (*Escherichia-Shigella*) were significantly enriched in the phyllosphere, whereas OTU_17 (*Xanthomonas*), OTU_9 (*Achromobacter*), *Sphingobium* (OTU_2), *Serratia* (OTU_3), and *Pantoea* (OTU_1) were significantly enriched in the rhizosphere (Figure 6A). Moreover, a differential analysis of the relative abundances in the rice microbiome of the orders to which these strains belong revealed that, as compared to the SynCom, Burkholderiales, Enterobacteriales, and Sphingomonadales were significantly enriched in the rhizosphere (Appendix A). A variance analysis between the rhizosphere and phyllosphere showed that Bacillales, Burkholderiales, Enterobacteriales, Neisseriales, Rhizobiales, and Sphingomonadales were significantly enriched in the rhizosphere (Appendix A). These results further validate that the migration of microorganisms from the soil to rice is not a random process but rather a selective process with specificity.

For maize, as compared to the SynCom, OTU_3 (Enterobacteriales), OTU_2 (Sphingomonadales), and OTU_6 (Pseudomonadales) were significantly enriched in the rhizosphere, whereas OTU_7 and OTU_254 (Pseudomonadales), OTU_1 (Enterobacteriales), and OTU_45 (Bacillales) were significantly enriched in the phyllosphere (Figure 6B). A variance analysis between the maize phyllosphere and rhizosphere revealed that *Pseudomonas* (OTU_7, OTU_254, OTU_13), *Bacillus* (OTU_14, OTU_12, OTU_45), *Chromobacterium* (OTU_5), *Pantoea* (OTU_1), *Xanthomonas* (OTU_17), and *Acinetobacter* (OTU_8) were significantly enriched in the phyllosphere, while *Achromobacter* (OTU_6), *Pseudomonas* (OTU_28), *Rhizobium* (OTU_18), *Achromobacter* (OTU_9), *Escherichia-Shigella* (OTU_4), *Sphingobium* (OTU_2), and *Serratia* (OTU_3) were significantly enriched in the rhizosphere (Figure 6B). A differential analysis of the relative abundances in the maize microbiome of the orders to which these strains belong, compared to those in the SynCom, showed that Burkholderiales and Enterobacteriales were significantly enriched in the rhizosphere, while Enterobacteriales was significantly enriched in the phyllosphere (Appendix A). A variance analysis between the rhizosphere and phyllosphere revealed that Burkholderiales and Sphingomonadales were significantly enriched in the rhizosphere (Appendix A). These results further validate that while there are similarities in the migration of microorganisms from the soil to rice and maize, there is also specificity in this selective process.

### 3.5. Rice and Maize Can Assemble Microbiota from SynCom to Promote Plant Growth

The SynCom can be selected by both rice and maize from the soil. We also found that, compared to plants grown in soil without the SynCom, rice grown in SynCom-containing soil showed significantly increased shoot length (Figure 7E) and plant length (Figure 7F), while maize grown in SynCom-containing soil exhibited significantly enhanced shoot weight (Figure 8C), fresh weight (Figure 8D), shoot length (Figure 8E), and plant length (Figure 8F). This result indicates that plants promote their own growth by selecting species from the SynCom in the soil and these species migrate to the rhizosphere or phyllosphere.

To further uncover which species play a role in promoting plant growth after migration, we analyzed the microbiomes in the phyllosphere and rhizosphere. A PCoA analysis revealed that there was no significant difference in the phyllosphere microbiome of rice grown in soil without the SynCom compared to that of rice grown in soil containing the SynCom (Appendix A; *p* = 0.208, R^2^ = 0.2064, PERMANOVA). However, the SynCom showed a greater impact on the rice rhizosphere microbiome (Figure 7B, *p* = 0.068, R^2^ = 0.2857, PERMANOVA). This suggests that the microbiome that migrated to the rhizosphere may play a role in promoting plant growth. A further variance analysis indicated that, compared to that of rice grown in soil without the SynCom, the rhizosphere of rice grown in soil containing the SynCom showed significant enrichment of the bacterial genera *Serratia*, *Sphingobium*, and *Achromobacter* (Figure 7A). Interestingly, OTU_2 (*Sphingobium*) and OTU_9 (*Achromobacter*) from the SynCom showed significant enrichment in the rhizosphere (Appendix A). These results suggest that rice may promote its own plant growth by selectively recruiting *Sphingobium* and *Achromobacter* from the SynCom to its rhizosphere.

For maize, we also found that there was no significant difference in the phyllosphere microbiome of maize grown in soil without the SynCom compared to that of maize grown in soil containing the SynCom (Appendix A; *p* = 0.194, R^2^ = 0.1631, PERMANOVA). However, the SynCom showed a greater impact on the maize rhizosphere microbiome (Figure 8B, *p* = 0.004, R^2^ =0.8685, PERMANOVA). This suggests that the rhizosphere microbiome of maize may play a role in promoting plant growth. A further variance analysis indicated that, compared to that of maize grown in soil without the SynCom, the rhizosphere of maize grown in soil containing the SynCom showed significant enrichment of the bacterial genera *Serratia*, *Sphingobium*, *Achromobacter*, *Acinetobacter*, *Escherichia-Shigella*, and *Pandoraea* (Figure 7A). Interestingly, OTU_3 (*Serratia*), OTU_8 and OTU_6 *(Acinetobacter*), OTU_4 (*Escherichia-Shigella*), and OTU_9 (*Achromobacter*) from the SynCom showed significant enrichment in the rhizosphere (Appendix A). In summary, these findings indicate that, compared to rice, maize can selectively recruit a greater number of microbes from SynCom-containing soil into its rhizosphere, leading to more pronounced growth-promoting effects in maize than in rice. Additionally, a strain of *Achromobacter* belonging to the order Burkholderiales within the SynCom may be specifically selected by both maize and rice and could promote plant growth. This finding also confirms that both maize and rice rhizospheres can become enriched with members of Burkholderiales from soil.

## 4. Discussion

Rice and maize can enrich their rhizospheres and phyllospheres with specific microorganisms from within the soil microbiome. Specifically, Burkholderiales and Bacteroidales, two groups within the soil microbiome, are enriched in the rhizosphere and phyllosphere, respectively, of both rice and maize. The enrichment of Burkholderiales in the rhizosphere of rice and maize cultivated in various soil environments might be associated with comparable recruitment mechanisms shared by the host plants. For instance, both rice and maize secrete flavonoid-like substances from their roots, which are capable of recruiting Burkholderiales [60]. Many studies have found that Bacteroidales are among the predominant species in the plant phyllosphere, a prevalence that may be attributed to their adaptability to the plant environment [54,61]. However, compared to the maize rhizosphere, the rice rhizosphere shows Caulobacterales enrichment. Currently, there is limited research on Caulobacterales colonization in the rice rhizosphere, but they are among the predominant species in the white clover rhizosphere, indicating the potential for Caulobacterales to become enriched in the rice rhizosphere [62]. During the migration of the soil microbiome to the phyllosphere, Pseudomonadales and Enterobacteriales are enriched in the rice phyllosphere, whereas Sphingomonadales are enriched in the maize phyllosphere. Soil serves as a critical reservoir for bacterial migration to the rhizosphere and phyllosphere [63]. Previously suggested mechanisms include soil bacterial migration toward the rhizosphere, via chemotaxis, cell motility, and quorum sensing, and toward the phyllosphere through surfaces, vessels, and gas convection [37,64]. Moreover, the environmental filtering and enrichment of specific microorganisms by the host represent important mechanisms that filter microbiome migration to different plant compartments [27]. Studies on the rice phyllosphere have indicated that Pseudomonadales and Enterobacteriales can accumulate in the rice phyllosphere, which may be related to specific selection mechanisms in the rice phyllosphere, such as the metabolite 4-HCA (4-hydroxycinnamic acid) specifically recruiting Pseudomonadales [54]. Recent studies have found that Sphingomonadales are enriched in the maize microbiome, indicated that maize also possesses specific selectivity for Sphingomonadales [65,66]. However, the specific recruitment mechanisms remain to be further elucidated.

Specific microbial functions within the soil microbiome can be enriched by rice and maize. Processes such as chemoheterotrophy and aerobic chemoheterotrophy are enriched in the rhizosphere microbiomes of both maize and rice. Conversely, functions including fermentation, animal parasites or symbionts, and nitrate reduction are enriched in the phyllosphere communities of both crops. In the rhizosphere of wild rice, maize, cowpea, and *Caragana korshinskii* grown in the field, there is also an enrichment of chemoheterotrophy and aerobic chemoheterotrophy functions [67,68,69,70,71]. Meanwhile, the phyllospheres of *Eucommia ulmoides*, maize, and cultivated rice grown in the field have been found to contain functionalities related to fermentation, nitrate reduction, and animal parasites or symbionts [72,73,74]. These results indicate that plants are capable of selecting microbes with these functionalities in both their rhizospheres and leaves. However, compared to rice, maize enriches more functionalities in its rhizosphere, such as ureolysis, nitrogen fixation, and hydrocarbon degradation. Conversely, the phyllosphere of rice is enriched with more arsenite oxidation detoxification and dissimilatory arsenite oxidation functionalities. The maize root system secretes mucilage that attracts nitrogen-fixing bacteria, and strains harboring nitrogen-fixing genes have been isolated from the maize rhizosphere [75,76]. Consequently, it is hypothesized that maize may possess a more robust capability to recruit nitrogen-fixing bacteria than rice. Arsenite oxidation detoxification and dissimilatory arsenite oxidation are notably enriched in the rice phyllosphere, which may be attributed to the specific enrichment of Pseudomonadales within this microenvironment. Previous research has demonstrated that bacteria belonging to the genus *Pseudomonas* can effectively oxidize, adsorb, and remove arsenite [77]. This suggests that Pseudomonadales present in the rice phyllosphere might play a crucial role in mitigating arsenic toxicity through these mechanisms.

Certain strains are selectively recruited from a SynCom to the rhizosphere and phyllosphere by both rice and maize. Notably, strains from the orders Burkholderiales and Sphingomonadales present in these SynComs are particularly enriched in the rhizospheres of both crops. The dominant microbial community in the rhizosphere of maize cultivated under various agricultural systems is primarily composed of members of the orders Sphingomonadales and Burkholderiales [65,78]. The rhizosphere of rice can also recruit strains from the orders Burkholderiales and Sphingomonadales for fungal resistance and plant growth promotion [79,80,81]. Strains from the orders Rhizobiales and Bacillales are enriched in the rhizosphere and phyllosphere of maize, respectively. Although a substantial body of research indicates that Bacillales primarily interact with plants in the rhizosphere, there is also evidence suggesting that Bacillales can be used to control diseases on leaves [82,83,84,85,86]. This leads to the speculation that maize may have the capability to recruit Bacillales to its phyllosphere. Field-grown maize can enrich nitrogen-fixing microorganisms such as Rhizobiales in its rhizosphere at different developmental stages to promote its growth [65,87]. Strains belonging to the orders Pseudomonadales and Enterobacteriales demonstrate both overlapping and distinct enrichment patterns within the rhizospheres and phyllospheres of maize and rice. Specifically, OTU_6 (Pseudomonadales) and OTU_3 (Enterobacteriales) are commonly enriched in the rhizospheres of both crops, whereas OTU_1 (Enterobacteriales) and OTU_4 (Enterobacteriales) show preferential enrichment in the rhizospheres of rice and maize, respectively. Members of the Pseudomonadales and Enterobacteriales orders engage in a variety of interaction patterns within the rhizospheres and phyllospheres of rice and maize [54,88,89,90,91,92]. These interactions are significantly shaped by the selective pressures exerted by their respective hosts. Therefore, colonization in the rhizosphere and phyllosphere may dynamically change with variations in the host phenotype. Further research is needed to dissect the molecular mechanisms of host–microbe interactions at the strain level.

Rice and maize attract specific microorganisms to colonize the rhizosphere, thereby promoting plant growth. Compared to rice, maize facilitates the migration of a greater number of microorganisms from SynCom to its rhizosphere for growth promotion. This may be attributed to the fact that maize seeds are directly exposed to the soil environment, allowing their surface microbes to colonize more easily during the initial germination stage. In contrast, rice seeds, enclosed by the hull, have limited early microbial contact [93]. However, rice and maize select *Achromobacter* from the Burkholderiales order in the soil to migrate to the rhizosphere to promote plant growth. *Achromobacter* has been isolated from maize, banana, halophytes such as Mesquite (*Prosopis* sp.), *Medicago sativa* L., and soil. This bacterium exhibits multiple functions including promoting plant growth, reducing pathogen infection rates, solubilizing phosphate, degrading soil contaminants (particularly polycyclic aromatic hydrocarbons, PAHs), enhancing salt stress tolerance, and degrading glyphosate [51,94,95,96,97]. Research has found that many genera in Burkholderiales, including *Achromobacter*, possess nitrogen-fixing capabilities [98]. These results indicate that *Achromobacter* is a plant-beneficial species that can be recruited from the soil by plants, and this beneficial effect may have broad-spectrum applicability.

## 5. Conclusions

Our study provides new insights into how soil microbiomes migrating towards the rhizosphere and phyllosphere are selectively assembled by plants to enhance their suitability. In our experiments, the composition and functions of the soil microbiome underwent significant changes as it migrated towards the rhizosphere and phyllosphere of maize and rice. Moreover, our data indicate that specific strains within the Burkholderiales order were selectively recruited from the SynCom to the rhizosphere by both rice and maize. Furthermore, rice and maize selectively recruited *Achromobacter* strain from the soil Burkholderiales community to the rhizosphere, which enhanced plant growth.

## Figures and Tables

**Figure 1 microorganisms-13-00947-f001:**
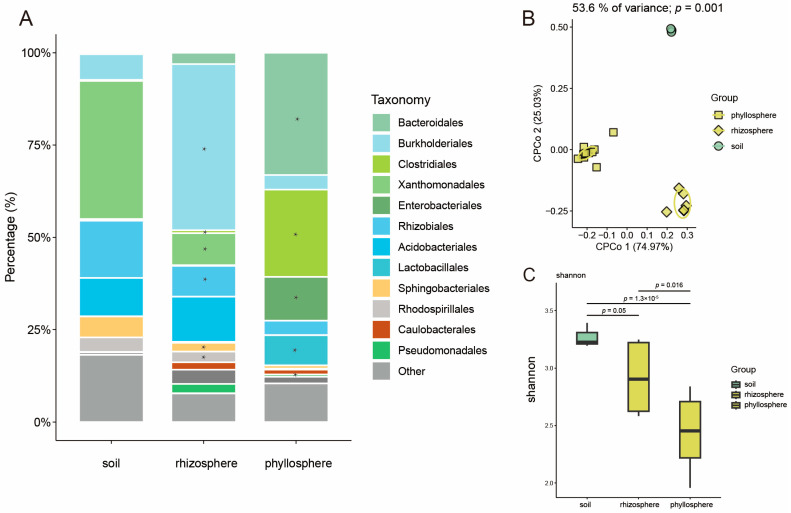
The microbiome analysis in different compartments of rice. (**A**) Order-level distribution of bacteria in rice microbiomes. The asterisks indicate that the relative abundances are significantly different from those in the soil (*p* < 0.05). (**B**) Constrained Principal Coordinate Analysis (CPCoA) of bacterial communities in the phyllosphere, rhizosphere, and soil of rice (*p*-value was calculated by one-way PERMANOVA). Ellipses cover 68% of the data for each group. (**C**) Shannon index for bacterial communities inhabiting soil, phyllosphere, and rhizosphere. The horizontal bars within boxes represent medians. The tops and bottoms of boxes represent the 75th and 25th percentiles, respectively. The upper and lower whiskers extend to data no more than 1.5× the interquartile range from the upper edge and lower edge of the box, respectively. The replications of samples in soil, rhizosphere, and phyllosphere are 3, 6, and 6, respectively. In this figure, the *p*-value is calculated with an unpaired two-sided *t*-test.

**Figure 2 microorganisms-13-00947-f002:**
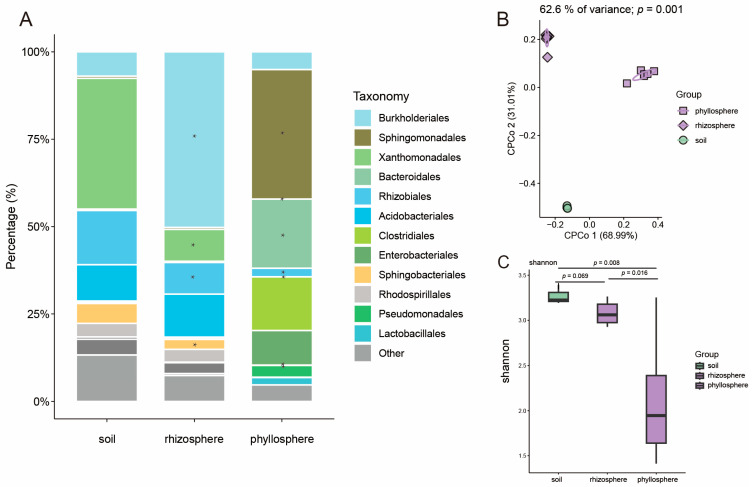
The microbiome analysis in different compartments of maize. (**A**) Order-level distribution of bacteria in maize microbiomes. The asterisks indicate that the relative abundances are significantly different from those in the soil. (**B**) CPCoA of bacterial communities in the phyllosphere, rhizosphere, and soil of maize (*p*-value was calculated by one-way PERMANOVA). Ellipses cover 68% of the data for each group. (**C**) Shannon index for bacterial communities inhabiting soil, phyllosphere, and rhizosphere. The replications of samples in the soil, rhizosphere, and phyllosphere are 3, 6, and 6, respectively. In this figure, the *p*-value is calculated with an unpaired two-sided *t*-test, and box plot percentiles are the same as in Figure 1.

**Figure 3 microorganisms-13-00947-f003:**
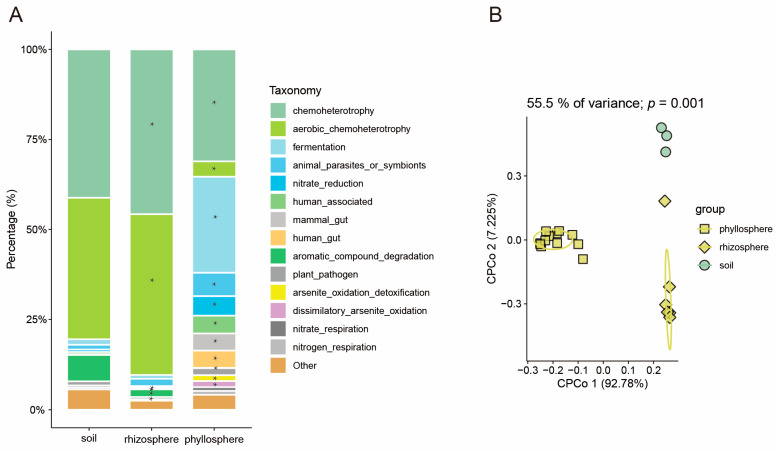
Microbiome function analysis in different compartments of rice. (**A**) Distribution of bacterial functions in rice. The asterisks indicate that the relative abundances are significantly different from those in the soil (*p* < 0.05). The *p*-value was calculated with an unpaired two-sided *t*-test. (**B**) CPCoA of bacterial functions in the phyllosphere, rhizosphere, and soil of rice (*p*-value was calculated by one-way PERMANOVA). Ellipses cover 68% of the data for each group.

**Figure 4 microorganisms-13-00947-f004:**
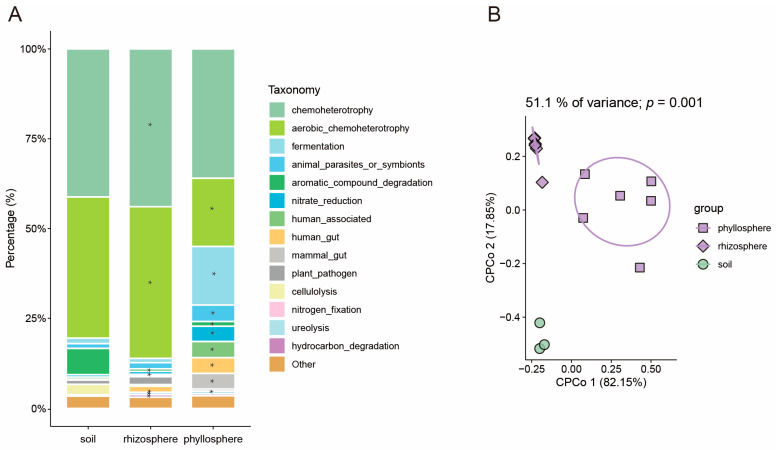
Microbiome function analysis in different compartments of maize. (**A**) Distribution of bacterial functions in maize. The asterisks indicate that the relative abundances are significantly different from those in the soil (*p* < 0.05). (**B**) CPCoA of bacterial functions in the phyllosphere, rhizosphere, and soil of maize (*p*-value was calculated by one-way PERMANOVA). Ellipses cover 68% of the data for each group.

**Figure 5 microorganisms-13-00947-f005:**
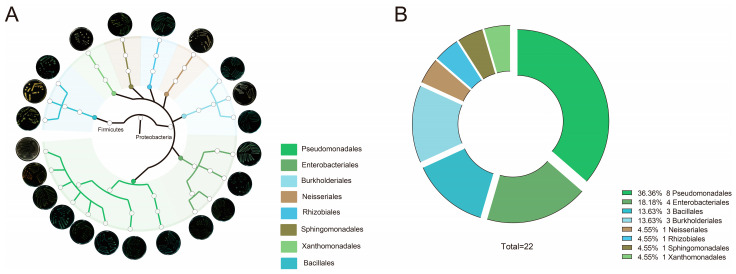
Classification and proportions of SynCom at the order-level. (**A**) A phylogenetic tree of 22 strains, where branches are color-coded to indicate different orders and the outermost images depict the phenotypic characteristics corresponding to each strain. (**B**) A pie chart that illustrates the compositional proportions of the 22 strains at the order level, with distinct colors representing various orders.

**Figure 6 microorganisms-13-00947-f006:**
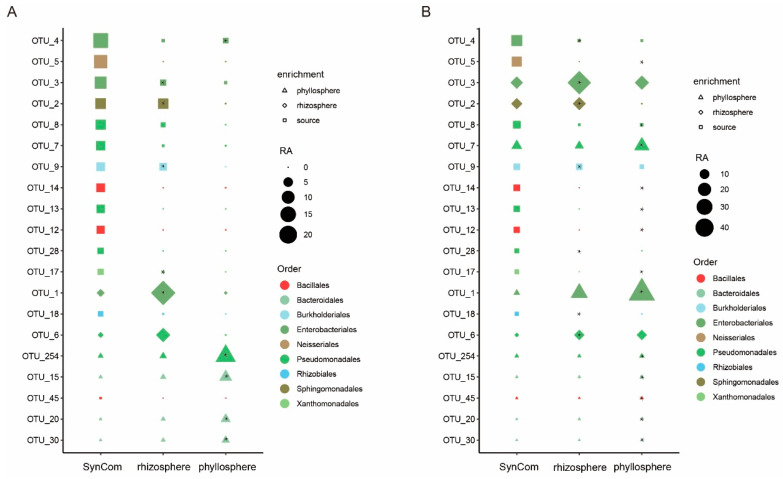
Heatmap of the abundance of the top 20 OTUs in the SynCom across different compartments. (**A**) Dynamic of the top 20 OTUs in the SynCom in rice; (**B**) Dynamic of the top 20 OTUs in the SynCom in maize. Triangles, diamonds, and squares indicate OTUs enriched in the phyllosphere, rhizosphere, and SynCom, respectively. The size of the symbols corresponds to the abundance level, and the color filling the symbols represents the order corresponding to the OTU; the asterisks indicate that the relative abundances are significantly different from SynCom (*p* < 0.05).

**Figure 7 microorganisms-13-00947-f007:**
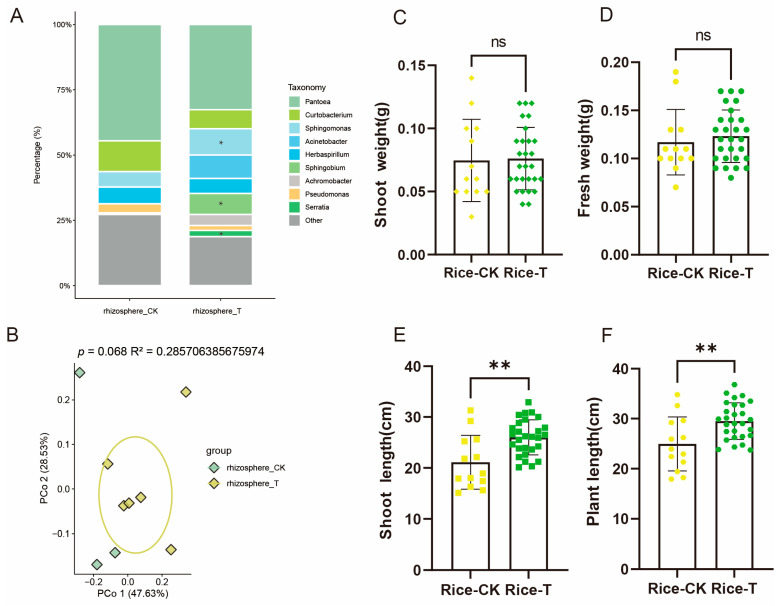
An analysis of the rhizosphere microbiota and phenotypic variance in rice plants inoculated and uninoculated with SynCom. (**A**) Genus-level distribution of bacteria in rice rhizosphere microbiome. (**B**) Unconstrained PCOA of bacterial communities in the rhizosphere of rice inoculated and uninoculated with SynCom (*p*-value was calculated by one-way PERMANOVA). Ellipses cover 68% of the data for each group. The differential analysis of shoot weight (**C**), fresh weight (**D**), shoot length (**E**), and plant length (**F**) between rice plants inoculated and non-inoculated with SynCom. Yellow symbols represent rice samples without SynCom inoculation, while green symbols represent rice samples with SynCom inoculation. Symbols of different shapes represent different agronomic traits of the rice samples. Specifically, (**C**) diamonds, (**D**) circles, (**E**), squares, and (**F**) hexagons represent shoot weight, fresh weight, shoot length, and plant length, respectively. The significance levels are as follows: ns (*p* > 0.05), * (*p* ≤ 0.05), and ** (*p* ≤ 0.01).

**Figure 8 microorganisms-13-00947-f008:**
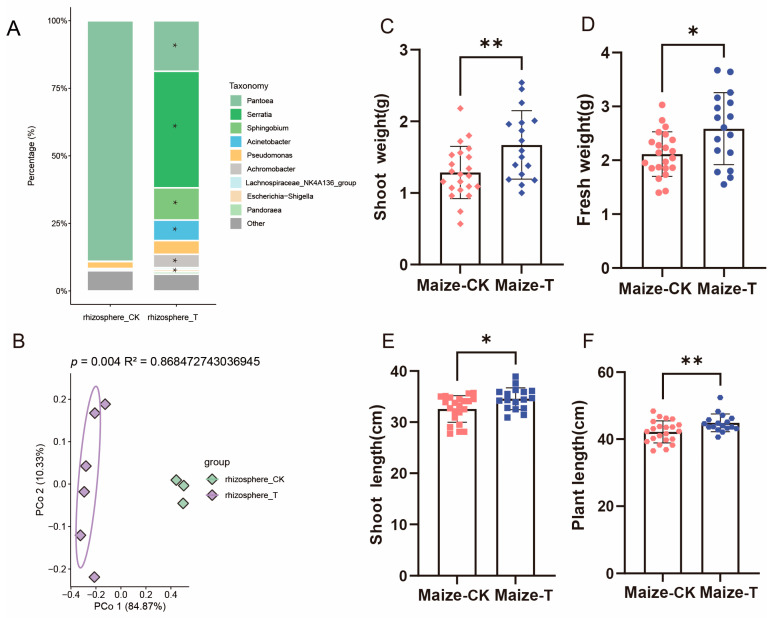
An analysis of the rhizosphere microbiota and phenotypic variance in maize plants inoculated and uninoculated with SynCom. (**A**) Genus-level distribution of bacteria in maize rhizosphere microbiome. (**B**) Unconstrained PCOA of bacterial communities in the rhizosphere of maize inoculated and uninoculated with SynCom (*p*-value was calculated by one-way PERMANOVA). Ellipses cover 68% of the data for each group. The difference analysis of shoot weight (**C**), fresh weight (**D**), shoot length (**E**), and plant length (**F**) between maize plants inoculated and non-inoculated with SynCom. Pink symbols represent maize samples without SynCom inoculation, while purple symbols represent maize samples with SynCom inoculation. Symbols of different shapes represent different agronomic traits of the maize samples. Specifically, (**C**) diamonds, (**D**) circles, (**E**) squares, and (**F**) hexagons represent shoot weight, fresh weight, shoot length, and plant length, respectively. The significance levels are as follows: * (*p* ≤ 0.05), and ** (*p* ≤ 0.01).

## Data Availability

Raw sequence data (16S rRNA gene fragment sequencing) generated in this study have been deposited in the Genome Sequence Archive of the BIG Data Center [122], Chinese Academy of Sciences under accession PRJCA035147 [https://ngdc.cncb.ac.cn/bioproject/browse/PRJCA035147].

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
