# Peer review of "Microbiome Migration from Soil to Leaves in Maize and Rice"

_microorganisms, 2025, doi:10.3390/microorganisms13040947_

Round 1

Reviewer 1 Report

Comments and Suggestions for Authors

This manuscript presents interesting results of studies on bacteria in soil and in roots and leaves of maize and rice. The aim of this study was to explain the influence of soil on the quantitative and qualitative composition of bacteria in roots and leaves, which the authors defined as ‘microbiome migration from soil to leaves’. The experiments were planned and carried out correctly from the methodological point of view. In addition, 22 bacterial strains isolated from the leaves of rice and maize were used to construct a synthetic microbial community (SynCom). The soil was inoculated with the SynCom and then it was determined how it affects the bacterial composition in the leaves of both tested plants. The results are presented in detail. Overall, these multi-threaded results are interesting and should be published in Microorganisms. However, some corrections should be made to the manuscript beforehand, which should take into account the comments presented in the manuscript text (see appendix). Unfortunately, the lines were not numbered, which made it difficult to add comments. In addition, in my opinion, changes should be made to certain concepts and terms used in the manuscript:

a/ the title should be changed. In this work, the migration of the microbiome was not studied. Also, in most cases, such a term should be removed from the text. If the authors believe that this term is correct, they should justify it in the Introduction. In addition, in the Discussion, the authors should present the mechanism by which such migration occurs.

b/ the authors use the term microbiome, but only bacteria were studied

c/ the authors use the term leaves = phyllosphere. Please note that these are not synonyms [the phyllosphere is the total above-ground surface of a plant when viewed as a habitat for microorganisms]

d/ the authors use the term roots = rhizosphere. Please note that these are not synonyms [the rhizosphere is the narrow region of soil or substrate that is directly influenced by root secretions and associated soil microorganisms]

Of course, it is permissible to specify in Material and Methods the meaning of terms used in a given work but they cannot be used as if they were synonyms.

Author Response

Main Comments and Suggestions:

Comments a. the title should be changed. In this work, the migration of the microbiome was not studied. Also, in most cases, such a term should be removed from the text. If the authors believe that this term is correct, they should justify it in the Introduction. In addition, in the Discussion, the authors should present the mechanism by which such migration occurs.

Response a: Thank you for pointing out this issue. We believe that the phrase ‘microbiome migration’ provides a more general description of the microbiome selected by plants in the title, which can enhance the article’s appeal and coherence. So, we keep the title in this manuscript. Additionally, we inserted a sentence in introduction section at line 68-70: “Soil serves as the primary source of the plant microbiome, and studies in Arabidopsis thaliana have demonstrated that soil-derived microbial communities can migrate to the phyllosphere.”

Moreover, we inserted three sentences in discussion section at line 482-488: “Soil serves as a critical reservoir for bacterial migration to the rhizosphere and phyllosphere. Previously suggested mechanisms include soil bacterial migration toward the rhizosphere via chemotaxis, cell motility, and quorum sensing, as well as toward the phyllosphere through surfaces, vessels, and gas convection. Moreover, the environmental filtering and enrichment of specific microorganisms by the host represent important mechanisms that filter the microbiome migrating to different plant compartments.”

Comments b. the authors use the term microbiome, but only bacteria were studied

Response b:This is a valid perspective. We have revised the descriptions in the Methods section and the figure legends. Since bacteria constitute the dominant group in the plant microbiome, and this terminology aligns better with existing research literature, we retain the use of 'microbiome' in the other parts.

Comments c. the authors use the term leaves = phyllosphere. Please note that these are not synonyms [the phyllosphere is the total above-ground surface of a plant when viewed as a habitat for microorganisms]

Response c: We have inserted additional descriptions in the Methods section (line116-120) to clarify ambiguity. Since leaves constitute the primary component of the phyllosphere, and this terminology aligns with standard nomenclature, we retain the use of “phyllosphere” throughout the manuscript.

Comments d. the authors use the term roots = rhizosphere. Please note that these are not synonyms [the rhizosphere is the narrow region of soil or substrate that is directly influenced by root secretions and associated soil microorganisms]. Of course, it is permissible to specify in Material and Methods the meaning of terms used in a given work but they cannot be used as if they were synonyms.

Response d: We have inserted additional descriptions in the Methods section (line 116-120) to clarify ambiguity. Additionally, we have replaced the term “root” with “rhizosphere” throughout the manuscript.

comments presented in the appendix text:

Comments 1.The important question is whether viruses belong to microbial communities?

Response 1: Pankaj Trivedi and colleagues (Nature reviews. Microbiology, 2020) concluded that “Plants provide a multitude of niches for the growth and proliferation of a diversity of microorganisms, including bacteria, fungi, protists, nematodes and viruses (the plant microbiota).”

Comments 2.The way of quoting should be standardized, in other places there is no space [I think there should always be a space]

Response 2: We have already made the necessary corrections in the manuscript by adding spaces between all citations and sentences.

Comments 3.“plant microbiome plays a cru cial role…” in this text is already written above (lines 5-7)

Response 3: We reshape as: “These research findings further underscore the importance of the plant microbiome in promoting plant growth and enhancing resistance to abiotic stresses.” at line 63-65.

Comments 4.number 47 is Zhou et al., not Wang et al.

Response 4: We have corrected it to: Zhou and colleagues (2022).

Comments 5.sterilization or surface sterilization

Response 5: We have revised “sterilization” to “surface sterilisation.”

Comments 6.rhizosphere and phyllosphere …why are such terms used and not roots and leaves

Response 6: We have inserted additional descriptions in the Methods section (line 116-120) to clarify ambiguity.

Comments 7.Only bacteria were studied, so the term microbiota is imprecise

Response 7: We reshape as: “Phyllosphere and rhizosphere microbiomes were extracted from these samples, then stored at -80℃ for further analysis of bacterial community composition” at line 183-185.

Comments 8.should be Fig. 1A (also in other places in the text)

Response 8: We have now revised all instances of figure citations in the manuscript, changing “Fig.” to “Figure”.

Comments 9.In Figure 1 only bacteria are shown and not the whole microbiome, at least there should be fungi

Response 9: We have revised the descriptions in the figure legends.

Comments 10.For maize, CPCoA and PERMANOVA showed a clear separation of microbial com munities between in maize.   not clear

Response 10: We sincerely appreciate you raising this issue. We have revised it to: “For maize, CPCoA and PERMANOVA showed a clear separation of microbial communities between different compartments of maize” at line 227-228.

Comments 11.dissimilatory arsenite oxidation (0.01%, 0.14%, 1.66%), nitrate respiration (0.52%, 0.08%, 1.06%) (Fig 2A).   probably Fig. 3A

Response 11: We have already corrected this citation and have checked all figure citations in this manuscript.

Comments 12.Figure 4. The Microbiome function analysis in different compartments of maize.   no citation in the text in Fig. 3B and Fig. 4B

Response 12: We have already corrected this citation and have checked all figure citations in this manuscript.

Comments 13.During the migration of soil microbiome to the phyllosphere, Pseudomonadales and Enterobacteriales are en riched in the rice phyllosphere, whereas Sphingomonadales are enriched in the maize phyllosphere. There must be a paragraph devoted to the discussion of the mechanism of migration. How does this process proceed according to the authors. Bacteria "flow" from the soil to the roots and then to the leaves?

Response 13: we inserted three sentences in discussion section at 482-487.

Comments 14.Caragana korshinskii, Pseudomonas in italic

Response 14: We have now used italic formatting for Caragana korshinskii and Pseudomonas, and have checked the formatting in this manuscript.

Reviewer 2 Report

Comments and Suggestions for Authors

The topic is intriguing to potential readers, and the manuscript presents relevant findings and information that respond to actual research questions. An example of this is the process by which plants selectively recruit microbiota from a shared soil source.

Overall, the manuscript is readable and interesting, but it could use some improvement. Furthermore, the manuscript contains several typos and grammatical mistakes that should be double-checked. Detailed suggestions are presented in the attached manuscript.

The main issue that I would like the authors to clarify is concern:

 - The introduction lacks a clearly defined objective and does not delve into different plant mechanisms (e.g., plant signals, chemotaxis, root exudate chemistry) in order to explain selective recruitment.

- SynCom characterisation is limited: 22 strains are isolated, but the rationale fo inclusion and their functional traits (e.g., PGP, stress tolerance, biosynthetic potential) are not discussed. In order to improve the manuscript, the authors could add a supplementary table showing known or predicted traits of the 22 strains.

- Conclusions are not mandatory but can be added to the manuscript in order to improve the interpretive depth of the results.

I suggest the authors improve the manuscript by adding the requested information in the evaluated manuscript and rephrasing some of the sentences in the attached manuscript. Furthermore, please rewrite the abstract and switch to passive voice, especially in the methods and results sections.

Comments on the Quality of English Language

Some minor improvements are needed. 

Author Response

Main Comments and Suggestions:

Comments a.The introduction lacks a clearly defined objective and does not delve into different plant mechanisms (e.g., plant signals, chemotaxis, root exudate chemistry) in order to explain selective recruitment.

Response a: We have added mechanisms that plant selectively recruit microorganisms in introduction at line 44-48: “Studies have demonstrated that plant rhizospheres secrete specific compounds, in-cluding flavonoids, citric acid, malic acid, and coumarins to recruit soil microorganisms. Following recruitment into the rhizosphere, a subset of these micro-organisms may colonize the phyllosphere under the selective pressure imposed by the plant host.” And line 68-70: “Soil serves as the primary source of the plant microbiome, and studies in Arabidopsis thaliana have demonstrated that soil-derived microbial communities can migrate to the phyllosphere”.

Comments b.SynCom characterisation is limited: 22 strains are isolated, but the rationale fo inclusion and their functional traits (e.g., PGP, stress tolerance, biosynthetic potential) are not discussed. In order to improve the manuscript, the authors could add a supplementary table showing known or predicted traits of the 22 strains.

Response b: We have added a supplementary table and relevant sentence to describe the functional traits of the 22 strains at line 335-337.

Comments c.Conclusions are not mandatory but can be added to the manuscript in order to improve the interpretive depth of the results.

Response c: We have added the conclusions section at line 561-569.

Comments d.I suggest the authors improve the manuscript by adding the requested information in the evaluated manuscript and rephrasing some of the sentences in the attached manuscript. Furthermore, please rewrite the abstract and switch to passive voice, especially in the methods.

Response d: We sincerely appreciate the valuable suggestions you provided for this manuscript. Regarding the issues mentioned in the manuscript, such as grammar errors, improper use of voice, word spelling, labeling of figures and tables, and reference citations, we have made every effort to improve and implemented the necessary revisions. As for the rewriting of the abstract that you mentioned, we have already made the corresponding changes.

Comments presented in the attached manuscript:

Comments 1.Please use passive construction...rephrase “we investigated....”

Response 1: We reshape as: “In this study, microbiome migration from soil to leaves in rice (Oryza sativa) and maize (Zea mays) was analyzed through 16S rRNA sequencing and phenotypic assessments” at line 12-14.

Comments 2.please use passive construction“we found that maize....”

Response 2: We reshape as: “We discovered that maize significantly enriched specific microbiota...” at line 14-15.

Comments 3.the entire abstract should be rewritten.“we found that....”

Response 3: We reshape as: “we discovered that strains belonging to Bacillales were enriched in the maize phyllosphere compared to the rice phyllosphere.” at line 19-20.

Comments 4.please rephrase “SynCom inoculation significantly promoted plant growth”

Response 4: We reshape as: “SynCom inoculation significantly enhanced plant growth.” at line 23

Comments 5.please reformulate...communities is plural...please correct

Response 5: We reshape as: “These microbial communities are referred to....” at line 34

Comments 6.please rephrase“It plays a crucial role in the promoting plants growth”

Response 6: We reshape as: “It plays a crucial role in promoting plant growth.” at line 36

Comments 7.Studies are mentioned, but is only one cited. Please correct! “Studies of soil exchange have shown that soil microorganisms colonize in plant phyllosphere”

Response 7: We reshape as: “A study on soil exchange showed that soil microorganisms colonize the plant phyllo-sphere.” at line 41-42.

Comments 8.please correct the grammatical mistake “A recent study have shown that beneficial....”

Response 8: We reshape as: “A recent study showed that beneficial microorganisms can induce systemic resistance in plants, thereby increasing plants’ disease resistance.” at line 57-58

Comments 9.please add genera or spp.“Streptomyces

Response 9: We have corrected it to: Streptomyces sp. SS8 at line 58.

Comments 10.add the year of the article“Zhang and colleagues.... ,Huang and colleagues.... ,Esau De la Vega Camarillo and colleagues.... ,Wang and colleagues.... ”

Response 10: We have added the year of the article at line 77-81.

Comments 11.sterilisation. please use british spelling throught the text

Response 11: We have corrected “sterilization” to “surface sterilization” at line 104.

Comments 12.19,000 g...please use spacing correctly

Response 12: It has been corrected in the manuscript at line 122.

Comments 13.Colours. Please use british spelling throught the text

Response 13: We have corrected “colors” to “colours” at line 155, 342.

Comments 14.Fig. or Figure? Please correct, because you used Figure 1 and not Fig.1 under graphics

Response 14: We have now corrected all instances of “Fig.” in the manuscript to “Figure” for consistency.

Round 2

Reviewer 2 Report

Comments and Suggestions for Authors

Dear Authors,

please use passive construction and eliminate all constructions which are starting with WE

Comments 2.please use passive construction“we found that maize....”

Response 2: We reshape as: “We discovered that maize significantly enriched specific microbiota...” at line 14-15.

Comments 3.the entire abstract should be rewritten.“we found that....”

Response 3: We reshape as: “we discovered that strains belonging to Bacillales were enriched in the maize phyllosphere compared to the rice phyllosphere.” at line 19-20.

Author Response

Main comment: please use passive construction and eliminate all constructions which are starting with WE.

Comments 1.please use passive construction“we found that maize....”

Response 1: We reshape as: “microbiota and functional traits were specifically enriched by maize in its phyllosphere and rhizosphere.” at line 14-15.

Comments 2.the entire abstract should be rewritten. “we found that....”

Response 2: We reshape as: “strains belonging to Bacillus were enriched in the maize phyllosphere compared to the rice phyllosphere.” at line 19-20. Additionally, we have revised this description throughout the manuscript.